# Causal Interpretation of Self-Attention in Pre-Trained Transformers

**Raanan Y. Rohekar**
Intel Labs
raanan.yehezkel@intel.com

**Yaniv Gurwicz**
Intel Labs
yaniv.gurwicz@intel.com

**Shami Nisimov**
Intel Labs
shami.nisimov@intel.com

## Abstract

We propose a causal interpretation of self-attention in the Transformer neural network architecture. We interpret self-attention as a mechanism that estimates a structural equation model for a given input sequence of symbols (tokens). The structural equation model can be interpreted, in turn, as a causal structure over the input symbols under the specific context of the input sequence. Importantly, this interpretation remains valid in the presence of latent confounders. Following this interpretation, we estimate conditional independence relations between input symbols by calculating partial correlations between their corresponding representations in the deepest attention layer. This enables learning the causal structure over an input sequence using existing constraint-based algorithms. In this sense, existing pre-trained Transformers can be utilized for zero-shot causal-discovery. We demonstrate this method by providing causal explanations for the outcomes of Transformers in two tasks: sentiment classification (NLP) and recommendation.

## 1 Introduction

Causality plays an important role in many sciences such as epidemiology, social sciences, and finance (25; 34). Understanding the underlying causal mechanisms is crucial for tasks such as explaining a phenomenon, predicting, and decision making. An automated discovery of causal structures from observed data alone is an important problem in artificial intelligence. It is particularity challenging when latent confounders may exist. One family of algorithms (35; 7; 8; 46; 30), called constraint-based, recovers in the large sample limit an equivalence class of the true underlying graph. However, they require a statistical test to be provided. The statistical test is often sensitive to small sample size, and in some cases it is not clear which statistical test is suitable for the data. Moreover, these method assume that data samples are generated from a single causal graph, and are designed to learn a single equivalence class. There are cases where data samples may be generated by different causal mechanisms and it is not clear how to learn the correct causal graph for each data sample. For example, decision process may be different among humans. Data collected about their actions may not be represented by a single causal graph. This is the case for recommender systems that are required to provide a personalized recommendation for a user, based on her past actions. In addition, it is desirable that such automated systems will provide a tangible explanation to why the specific recommendation was given.

Recently, deep neural networks, based on the Transformer architecture (39) have been shown to achieve state-of-the-art accuracy in domains such as natural language processing (9; 3; 45), vision (10; 19), and recommender systems (36; 18). The Transformer is based on the attention mechanism, where it calculates context-dependent weights for a given input (32). Given an input consisting of a sequence of symbols (tokens), the Transformer is able to capture complex dependencies, but it is unclear how they are represented in the Transformer nor how to extract them.

---

*All authors contributed equally.

37th Conference on Neural Information Processing Systems (NeurIPS 2023).

In this paper we bridge between structural causal models and attention mechanism used in the Transformer architecture. We show that self-attention is a mechanism that estimates a linear structural equation model in the deepest layer for each input sequence independently, and show that it represents a causal structure over the symbols in the input sequence. In addition, we show that an equivalence class of the causal structure over the input can be learned solely from the Transformer's estimated attention matrix. This enables learning the causal structure over a single input sequence, using existing constraint-based algorithms, and utilizing existing pre-trained Transformer models for zero-shot causal discovery. We demonstrate this method by providing causal explanations for the outcomes of Transformers in two tasks: sentiment classification (NLP) and recommendation.

## 2  Related Work

In recent years, several advances in causal reasoning using deep neural networks were presented. One line of work is causal modeling with neural networks (42; 12). Here, a neural network architecture follows a given causal graph structure. This neural network architecture is constructed to model the joint distribution over observed variables (12) or to explicitly learn the deterministic functions of a structural causal model (42). It was recently shown that a neural network architecture that explicitly learns the deterministic functions of an SCM can be used for answering interventional and counterfactual queries (43). That is, inference in rungs 2 and 3 in the ladder of causation (26), also called layers 2 and 3 in the Pearl causal hierarchy (2). Nevertheless, this architecture requires knowing the causal graph structure before training, and supports a single causal graph.

In another line of work, an attention mechanism is added before training, and a causal graph is inferred from the attention values at inference (21; 17). Attention values are compared against a threshold, and high values are treated as indication for a potential causal relation. Nauta et al. (21) validate these potential causal relations by permuting the values of the potential cause and measuring the effect. However, this method is suitable only if a single causal model underlay the dataset.

## 3  Preliminaries

First, we provide preliminaries for structural causal models and the attention mechanism in the Transformer-based neural architecture. Throughout the paper we denote vectors and sets with bold-italic upper-case (e.g., $\boldsymbol{V}$), matrices with bold upper-case (e.g., $\mathbf{A}$), random variables with italic upper-case (e.g., $X$), and models in calligraphic font (e.g., $\mathcal{M}$). Different sets of letters are used when referring to structural causal models and attention (see Appendix A-Table 1 for a list).

### 3.1  Structural Causal Models

A structural causal model (SCM) represents causal mechanisms in a domain (24; 35; 28). An SCM is a tuple $\{\boldsymbol{U}, \boldsymbol{X}, \mathcal{F}, P(\boldsymbol{U})\}$, where $\boldsymbol{U} = \{U_1, \dots, U_m\}$ is a set of latent exogenous random variables, $\boldsymbol{X} = \{X_1, \dots, X_n\}$ is a set of endogenous random variables, $\mathcal{F} = \{f_1, \dots, f_n\}$ is a set of deterministic functions describing the values $\boldsymbol{X}$ given their direct causes, and $P(\boldsymbol{U})$ is the distribution over $\boldsymbol{U}$. Each endogenous variable $X_i$ has exactly one unique exogenous cause $U_i$ ($m = n$). The value of an endogenous variable $X_i, \forall i \in [1, \dots, n]$ is

$$X_i \leftarrow f_i(\boldsymbol{Pa}_i, U_i) \tag{1}$$

(the left-arrow indicates an assignment due to cause-effect relation), where $\boldsymbol{Pa}_i$ is the set of direct causes (parents in the causal graph) of $X_i$. A graph $\mathcal{G}$ corresponding to the SCM consists of a node for each variable, and a directed edge for each direct cause-and-effect relation, as evident from $\mathcal{F}$. If the graph is directed and acyclic (DAG) then the SCM is called *recursive*.

In a special subclass of SCMs, referred in this paper linear-Gaussian SCM, a node value is a linear combination of its parents, and exogenous nodes are independently and normally distributed additive noise. Let $\mathbf{G}$ be a weight matrix, where $\mathbf{G}(i, j)$ is the weight of parent node $X_j$ determining the child node $X_i$ and $\mathbf{G}(i, k) = 0$ if $X_k$ is not a parent of $X_i$, and $\boldsymbol{\Lambda}$ be a diagonal matrix, where $\boldsymbol{\Lambda}(i, i)$ is the weight of exogenous node $U_i$. Then, $\forall i \in [1, \dots, n]$

$$X_i \leftarrow \mathbf{G}(i, \cdot)\boldsymbol{X} + \boldsymbol{\Lambda}(i, i)U_i, \tag{2}$$

and in matrix form $\boldsymbol{X} = \mathbf{G}\boldsymbol{X} + \boldsymbol{\Lambda}\boldsymbol{U}$. In the case of a recursive SCM, with nodes sorted topologically (ancestors before their descendants), the weight matrix $\mathbf{G}$ is strictly lower triangular (zeros on the diagonal). In this paper we learn the causal graph, described by the non-zero elements of $\mathbf{G}$, using constraint-based causal discovery approach, which generally requires assuming the following.

**Definition 1** (Causal Markov). *In a causally Markov graph, a variable is independent of all other variables, except its effects, conditional on all its direct causes.*

**Definition 2** (Faithfulness). *A distribution is faithful to a graph if and only if every independence relation true in the distribution is entailed by the graph.*

## 3.2 Self-Attention Mechanism

Attention is a mechanism that predicts context dependent weights (32) for the input sequence. In this paper we analyse the attention mechanism used by Vaswani et al. (39). Here, the input is a matrix $\mathbf{Y}$ where row vector $\mathbf{Y}(i, \cdot)$ is the embedding of symbol $s_i$ in an input sequence $\boldsymbol{s} = [s_1, \ldots, s_n]$. The attention matrix is calculated by $\mathbf{A} = softmax(\mathbf{Y}\mathbf{W}_{QK}\mathbf{Y}^\top)$, such that the rows of $\mathbf{A}$ sum to $1$[1]. Values matrix is computed by $\mathbf{V} = \mathbf{Y}\mathbf{W}_V$, where row $\mathbf{V}(i, \cdot)$ is the value vector of input symbol $s_i$. We treat $\mathbf{V}$ as a set of projection of $\mathbf{Y}$ on column vectors $\mathbf{W}_V$. Finally, the output embeddings are $\mathbf{Z} = \mathbf{A}\mathbf{V}$, where the $i$-th output embedding $\boldsymbol{Z}_i$ is the $i$-th row of $\mathbf{Z}$.

# 4 A Link between Pre-trained Self-Attention and a Causal Graph underlying an Input Sequence of Symbols

In this section we first describe self-attention as a mechanism that encodes correlations between symbols in an unknown structural causal model (Section 4.1). After establishing this relation, we present a method for recovering an equivalence class of the causal graph underlying the input sequence (Section 4.2). We also extend the results to the multi-head and multi-layer architecture (Section 4.3).

## 4.1 Self-Attention as Correlations between Nodes in a Structural Causal Model

We now describe how self-attention can be viewed as a mechanism that estimates the values of observed nodes of an SCM. We first show that the covariance over the outputs of self-attention is similar to the covariance over observed nodes of an SCM. Specifically, we model relations between symbols in an input sequence using a linear-Gaussian SCM at the output of the attention layer. In an SCM, the values over endogenous nodes, in matrix form, are $\boldsymbol{X} = \mathbf{G}\boldsymbol{X} + \boldsymbol{\Lambda}\boldsymbol{U}$, which means

$$\boldsymbol{X} = (\mathbf{I} - \mathbf{G})^{-1}\boldsymbol{\Lambda}\boldsymbol{U}. \tag{3}$$

Since $\mathbf{G}$ is a strictly lower-triangular weight matrix, $(\mathbf{I} - \mathbf{G})^{-1}$ is a lower unitriangular matrix, which is equal to the sum of a geometric series

$$(\mathbf{I} - \mathbf{G})^{-1} = \sum_{k=0}^{n-1} \mathbf{G}^k. \tag{4}$$

Thus, the $(i, j)$ element represents the sum of directed-paths' weights, for all directed paths from $X_j$ to $X_i$ having length up to $n - 1$. The weight of a directed path is the product of the weights of edges along the path. In case some of the nodes are latent confounders, then the matrix in Equation 4 may not be triangular, due to spurious associations. Equation 3 represents a system with input $\boldsymbol{U}$, output $\boldsymbol{X}$ and weights $(\mathbf{I} - \mathbf{G})^{-1}\boldsymbol{\Lambda}$. The covariance matrix of the output is

$$\begin{aligned}
\mathbf{C}_{\boldsymbol{X}} &= \mathbb{E}[(\boldsymbol{X} - \boldsymbol{\mu}_{\boldsymbol{X}})(\boldsymbol{X} - \boldsymbol{\mu}_{\boldsymbol{X}})^\top] = \\
&= \mathbb{E}[(\mathbf{I} - \mathbf{G})^{-1}\boldsymbol{\Lambda}(\boldsymbol{U} - \boldsymbol{\mu}_{\boldsymbol{U}})(\boldsymbol{U} - \boldsymbol{\mu}_{\boldsymbol{U}})^\top((\mathbf{I} - \mathbf{G})^{-1}\boldsymbol{\Lambda})^\top)] = \\
&= (\mathbf{I} - \mathbf{G})^{-1}\boldsymbol{\Lambda}\mathbb{E}[(\boldsymbol{U} - \boldsymbol{\mu}_{\boldsymbol{U}})(\boldsymbol{U} - \boldsymbol{\mu}_{\boldsymbol{U}})^\top]((\mathbf{I} - \mathbf{G})^{-1}\boldsymbol{\Lambda})^\top = \\
&= ((\mathbf{I} - \mathbf{G})^{-1}\boldsymbol{\Lambda})\mathbf{C}_{\boldsymbol{U}}((\mathbf{I} - \mathbf{G})^{-1}\boldsymbol{\Lambda})^\top,
\end{aligned} \tag{5}$$

where $\boldsymbol{\mu}_{\boldsymbol{X}} = (\mathbf{I} - \mathbf{G})^{-1}\boldsymbol{\Lambda}\boldsymbol{\mu}_{\boldsymbol{U}}$, and $\mathbf{C}_{\boldsymbol{U}}$ is the covariance matrix of exogenous variables $\boldsymbol{U}$.

---

[1]In the paper by Vaswani et al. (39) the weight matrix is $\mathbf{W}_{QK} = \mathbf{w}_Q\mathbf{w}_K^\top/\sqrt{d_K}$, where the weight matrices $\mathbf{W}_Q$ and $\mathbf{W}_K$ are learned explicitly and $d_K$ is the number of columns in these matrices.

An attention layer in a Transformer architecture, estimates an attention matrix $\mathbf{A}$ and a values matrix $\mathbf{V}$ from embeddings $\mathbf{Y}$ of an input sequence of symbols $s = [s_1, \ldots, s_n]^\top$. The output embeddings is calculated by $\mathbf{Z} = \mathbf{A}\mathbf{V}$. It is important to note that during self-attention training, an inductive bias is used in the form of learning separate representations for the attention matrix $\mathbf{A}$ and values matrix $\mathbf{V}$. It can be seen that the $j$-th element of the $i$-th row-vector, $\mathbf{V}(i,j)$ is the projection of the symbol $s_i$ input embedding $\mathbf{Y}(i, \cdot)$ on column vector $\mathbf{W}_V(\cdot, j)$. In other words, an embedding is projected on the direction of vector $\mathbf{W}_V(\cdot, j)$. Thus, each column of $\mathbf{V}$ can be seen as a projection of the input embeddings. Moreover, the attention matrix is shared with all projections for calculating the output embeddings. That is, for the $j$-th projection the output embedding is

$$\mathbf{Z}(\cdot, j) = \mathbf{A}\mathbf{V}(\cdot, j). \tag{6}$$

Denote $\boldsymbol{Z}_j \equiv \mathbf{Z}(\cdot, j)$ and $\boldsymbol{V}_j \equiv \mathbf{V}(\cdot, j)$ corresponding to projection $j$. Covariance of $\boldsymbol{Z}_j$ is

$$\begin{aligned}
\mathbf{C}_{\boldsymbol{Z}_j} &= \mathbb{E}[(\boldsymbol{Z}_j - \boldsymbol{\mu}_{\boldsymbol{Z}_j})(\boldsymbol{Z}_j - \boldsymbol{\mu}_{\boldsymbol{Z}_j})^\top] \\
&= \mathbf{A}\mathbb{E}[(\boldsymbol{V}_j - \boldsymbol{\mu}_{\boldsymbol{V}_j})(\boldsymbol{V}_j - \boldsymbol{\mu}_{\boldsymbol{V}_j})^\top]\mathbf{A}^\top = \\
&= \mathbf{A}\mathbf{C}_{\boldsymbol{V}_j}\mathbf{A}^\top,
\end{aligned} \tag{7}$$

where $\mathbf{C}_{\boldsymbol{V}_j}$ is the covariance matrix of input embeddings projected on vector $\mathbf{W}_V(\cdot, j)$. Since columns of $\mathbf{V}$ describe different projections corresponding to independent columns of $\mathbf{W}_V$, and from the central limit theorem it follows that $\mathbf{C}_{\mathbf{V}} \to \mathbf{I}$, which leads to

$$\mathbf{C}_{\mathbf{Z}} = \mathbf{A}\mathbf{A}^\top. \tag{8}$$

How does attention matrix $\mathbf{A}$ and a values vector $\boldsymbol{V}_j$ translate into a structural causal model? We interpret the attention matrix $\mathbf{A}$ and values vector $\boldsymbol{V}$ as components of an SCM, such that the covariance matrix of its endogenous variables is equal to the covariance matrix of the attention layer's output, $\mathbf{C}_{\boldsymbol{X}} = \mathbf{C}_{\mathbf{Z}}$. From Equation 5 and Equation 8,

$$\mathbf{A}\mathbf{A}^\top = ((\mathbf{I} - \mathbf{G})^{-1}\boldsymbol{\Lambda})\mathbf{C}_{\boldsymbol{U}}((\mathbf{I} - \mathbf{G})^{-1}\boldsymbol{\Lambda})^\top. \tag{9}$$

The attention matrix $\mathbf{A}$ measures the total effect each variable has on another variable, for any projection $\mathbf{V}(\cdot, j)$ of the input embeddings. Similarly, matrix $(\mathbf{I} - \mathbf{G})^{-1}\boldsymbol{\Lambda}$ in an SCM measures the effect one variable has on another variable through all directed paths in the causal graph (Equation 4) for any value of the exogenous variables $\boldsymbol{U}$. Thus, form Equation 9 matrix $\mathbf{A}$ is analogous to $(\mathbf{I}-\mathbf{G})^{-1}\boldsymbol{\Lambda}$ and $\mathbf{V}(\cdot, j)$ to value assignment for $\boldsymbol{U}$, which serves as context. Both $\mathbf{A}$ and $(\mathbf{I}-\mathbf{G})^{-1}\boldsymbol{\Lambda}$ are not required to be triangular as latent confounders may introduce spurious associations, such that variables might be required to estimate the values of variables earlier in the topological order. In addition, the symbols in the sequence $\boldsymbol{S}$ are not assumed to be topologically ordered.

## 4.2 Attention-based Causal-Discovery (ABCD)

In Section 4.1 we show that self-attention learns to represent pairwise associations between symbols of an input sequence, for which there exists a linear-Gaussian SCM that has the exact same pair-wise associations between its nodes. However, can we recover the causal structure $\mathcal{G}$ of the SCM solely from the weights of a pre-trained attention layer? We now present the Attention-Based Causal-Discovery (ABCD) method for learning an equivalence class of the causal graph that underlies a given input sequence.

Causal discovery (causal structure learning) from observed data alone requires placing certain assumptions. Here we assume the causal Markov (24) and faithfulness (35) assumptions. Under these assumptions, constraint-based methods use tests of conditional independence (CI-tests) to learn the causal structure (27; 35; 8; 7; 30; 31; 22). A statistical CI-test is used for deciding if two variables are statistically independent conditioned on a set of variables. Commonly, partial correlation is used for CI-testing between continuous, normally distributed variables with linear relations. This test requires only a pair-wise correlation matrix (marginal dependence) for evaluating partial correlations (conditional dependence). We evaluate the correlation matrix from the attention matrix. From Equation 8 the covariance matrix of output embeddings is $\mathbf{C}_{\mathbf{Z}} = \mathbf{A}\mathbf{A}^\top$, and (pair-wise) correlation coefficients are $\rho_{i,j} = \mathbf{C}_{\mathbf{Z}}(i,j)/\sqrt{\mathbf{C}_{\mathbf{Z}}(i,i)\mathbf{C}_{\mathbf{Z}}(j,j)}$. Unlike kernel-based CI tests (1; 11; 13; 14; 37; 48), we do not need to explicitly define or estimate the kernel, as it is readily

available by a single forward-pass of the input sequence in the Transformer. This implies the following. Firstly, our CI-testing function is inherently learned during the training stage of a Transformer, by that enjoying the efficiency in learning complex models from large datasets. Secondly, since attention is computed for each input sequence uniquely, CI-testing is unique to that specific sequence. That is, conditional independence is tested under the specific context of the input sequence.

Finally, the learned causal graph represents an equivalence class in the form of a partial ancestral graph (PAG) (29; 47), which can also encode the presence of latent confounders. A PAG represents a set of causal graphs that cannot be refuted given the data. There are three types of edge-marks (at some node $X$): an arrow-head '$\longrightarrow X$', a tail '$\longrightarrow X$', and circle '$\longrightarrow$o $X$' which represent an edge-mark that cannot be determined given the data. Note that reasoning from a PAG is consistent with every member in the equivalence class it represents.

What is the relation between the equivalence class learned by ABCD and the causal graph underlying the input sequence $\boldsymbol{s} = [s_1, \ldots, s_n]$?

**Definition 3** (SCM-consistent probability-space). *Let $\Omega_n = \{[s_1, \ldots, s_n] : s_i \in \mathcal{S}\}$ be a sample space, where $\mathcal{S}$ is a countable set of symbols. Let $\mathcal{F}_n$ be an event space, and $P_n$ be a probability measure. For a finite $n \in \mathbb{N}$, the probability space $(\Omega_n, \mathcal{F}_n, P_n)$ is called SCM-consistent if every sequence $\boldsymbol{s} \in \Omega_n$, was generated by a corresponding SCM with graph $\mathcal{G}_{\boldsymbol{s}}$.*

**Assumption 1.** *Attention mechanism $\mathcal{M}$ is trained on sequences sampled from an SCM-consistent probability space, such that each input symbol is uniquely represented by its output embedding.*

Many training methods, such as MLM (9), satisfy Assumption 1 as they train the network to predict input symbols using their corresponding output embeddings in the deepest layer. This assumption is required for relating the causal graph learned from the deepest attention layer to the causal graph underlying the input sequence.

**Theorem 1.** *If $\boldsymbol{s}$ is a sequence from an SCM-consistent probability space used to train $\mathcal{M}$, $\mathcal{G}_{\boldsymbol{s}}$ is the causal graph of the SCM from which $\boldsymbol{s}$ was generated, and $\mathbf{A}$ the attention matrix for input sequence $\boldsymbol{s}$ in the deepest layer, then $\mathcal{G}_{\boldsymbol{s}}$ is in the equivalence class learned by a sound constraint-based algorithm that uses covariance matrix $\mathbf{C_Z} = \mathbf{A}\mathbf{A}^\top$ for partial correlation CI-testing.*

**Corollary 1.** *Nodes $X_i$ and $X_j$ are d-separated in $\mathcal{G_Z}$ conditioned on $\boldsymbol{X}' \subset \boldsymbol{X}$ if and only if symbols $s_i$ and $s_j$ are d-separated in $\mathcal{G}_{\boldsymbol{s}}$ conditioned on $\boldsymbol{s}'$, a subset containing symbols corresponding to $\boldsymbol{X}'$.*

Corollary 1 is useful for cases in which the graph over the input symbols $\mathcal{G}_{\boldsymbol{s}}$, as well as the structural equation model described by $\mathcal{G_Z}$, does not have a causal meaning. In these cases, one can reason about conditional independence relations among input symbols.

### 4.3 Multiple Attention Layers and Multi-Head Attention as a Structural Causal Model

Commonly, the encoder module in the Transformer architecture (39) consists of multiple self attention layers executed sequentially. A non-linear transformation is applied to the the output embeddings of one attention layer before feeding it as input to the next attention layer. The non-linear transformation is applied to each symbol independently. In addition each attention layer consists of multiple self-attention heads, processing the input in parallel using head-specific attention matrix. The output embedding of each symbol is then linearly combined along the heads. It is important to note that embedding of each symbol is processed independently of embeddings of other symbols. The only part in which an embedding of one symbol is influenced by embeddings of other symbol is the multiplication by the attentnion matrix.

Thus, a multi-layer, multi-head, architecture can be viewed as a deep graphical model, where exogenous nodes of an SCM are estimated by a previous attention layer. The different heads in an attention layer learn different graphs for the same input, where their output embeddings are linearly combined. This can be viewed as a mixture model. We recover the causal graph from the last (deepest) attention layer, and treat earlier layers as context estimation, which is encoded in the values of the exogenous nodes (Figure 1).

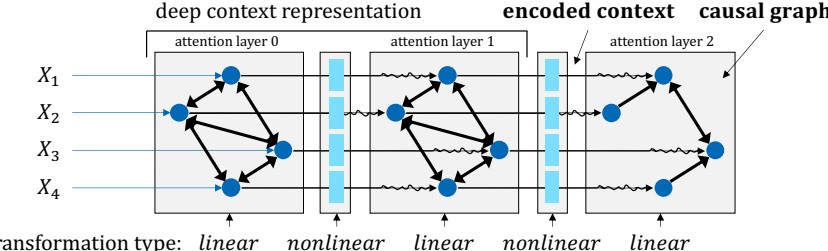

Figure 1: A network with three attention layers. Early attention layers (0 and 1) estimate context, which is encoded in the values of exogenous nodes for the last layer (2).

## 4.4 Limitations of ABCD

The presented method, ABCD, is susceptible to two sources of error affecting its accuracy. The first is prediction errors of the Transformer neural network, and the second is errors from the constraint-based causal discovery algorithm that uses the covariance matrix calculated in the Transformer. The presence of errors from the second source depends on the first. For a Transformer with perfect generalization accuracy, and when causal Markov and faithfulness assumptions are not violated, the second source of errors vanishes, and a perfect result is returned by the presented ABCD method.

From Assumption 1, the deepest attention layer is expected to produce embeddings for each input symbol, such that the input symbol can be correctly predicted from its corresponding embedding (one-to-one embedding). However, except for trivial cases, generalization accuracy is not perfect. Thus, the attention matrix of an input sequence, for which the Transformer fails to predict the desired outcome, might include errors. As a result, the values of correlation coefficients calculated from the attention matrix might include errors.

Constraint-based algorithms for causal discovery are generally proved to be sound and complete when a perfect CI-test is used. However, if a CI-test returns an erroneous result, the constraint-based algorithm might introduce additional errors. This notion is called stability (35), which is informally measured by the number of output errors as a function of the number of CI-tests errors. In fact, constraint-based algorithm differ from one another in their ability to handle CI-test errors and minimize their effect. Thus, inaccurate correlation coefficients used for CI-testing might lead to erroneous independence relations, which in turn may lead to errors in the output graph.

We expect that as Transformer models become more accurate with larger training datasets, the accuracy of ABCD will increase. In this paper's experiments we use existing pre-trained Transformers and common datasets, and use the ICD algorithm (30) that was shown to have state-of-the-art accuracy.

## 5 ABCD with Application To Explaining Predictions

One possible application of the proposed ABCD approach is to reason about predictions by generating causal explanations from pre-trained self-attention models such as BERT (9). Specifically, we seek an explaining set that consists of a subset of the input symbols that are claimed to be solely responsible for the prediction. That is, if the effect of the explaining set on the prediction is masked, a different (alternative) prediction is provided by the neural network.

It was previously claimed that attention cannot be used for explanation (16); however, in a contradicting paper (41), it was shown that explainability is task dependent. A common approach is to use the attention matrix to learn about input-output relations for providing explanations (33; 5; 4). These often rely on the assumption that inputs having high attention values with the class token influence the output (44; 6). We claim that this assumption considers only marginal statistical dependence and ignores conditional independence and explaining-away that may arise due to latent confounders.

To this end, we propose CLEANN (**C**ausa**L** **E**xplanations from **A**ttention in **N**eural **N**etworks). A description of this algorithm is detailed in Appendix B, and an overview with application to explaining movie recommendation is given in Figure 4. See also Nisimov et al. (23) for more details.

# 6 Empirical Evaluation

In this section we demonstrate how a causal graph constructed from a self-attention matrix in a Transformer based model can be used to explain which specific symbols in an input sequence are the causes of the Transformer output. We experiment on the tasks of sentiment classification, which classifies an input sequence, and recommendation systems, which generates a candidate list of recommended symbols (top-$k$) for the next item. For both experiments in this section we compare our method against two baselines from (38): (1) *Pure-Attention* algorithm (Pure-Atten.), that uses the attention weights directly in a hill-climbing search to suggest an explaining set, (2) *Smart-Attention* algorithm (Smart-Attn.), that adapts *Pure-Attention*, where a symbol is added to the explanation only if it reduces the score gap between the original prediction's score and the second-ranked prediction's score. Implementation tools are in `https://github.com/IntelLabs/causality-lab`.

## 6.1 Evaluation metrics

We evaluate the quality of inferred explanations using two metrics. One measures minimality of the explanation, and the other measures how specific is the explanation to the prediction.

### 6.1.1 Minimal explaining set

An important requirement is having the explaining set minimal in the number of symbols (40). The reason for that is twofold. (1) It is more complicated and less interpretable for humans to grasp the interactions and interplay in a set that contains many explaining symbols. (2) In the spirit of occum's razor, the explaining set should not include symbols that do not contribute to explaining the prediction (when faced with a few possibles explanations, the simpler one is the one most likely to be true). Including redundant symbols in the explaining set might result in a wrong alternative predictions when the effect of this set is masked.

### 6.1.2 Influence of the explaining set on replacement prediction

The following metric is applicable only to tasks that produce multiple predictions, such as multiple recommendations by a recommender system (e.g., which movie to watch next), as opposed to binary classification (e.g., sentiment) of an input. Given an input sequence consisting of symbols, $s = \{s_1, \ldots, s_n\}$, a neural network suggests $\tilde{s}_{n+1}$, the 1st symbol in the top-$k$ candidate list. CLEANN finds the smallest explaining set within $s$ that influenced the selection of $\tilde{s}_{n+1}$. As a consequence, discarding this explaining set from that sequence should prevent that $\tilde{s}_{n+1}$ from being selected again, and instead a new symbol should be selected in replacement (replacement symbol). Optimally, the explaining set should influence the rank of only that 1st symbol (should be downgraded), but not the ranks of the other candidates in the top-$k$ list. This requirement implies that the explaining set is unique and important for the isolation and counterfactual explanation of only that 1st symbol, whereas the other symbols in the original top-$k$ list remain unaffected, for the most part. It is therefore desirable that after discarding the explaining set from the sequence, the new replacement symbol would be one of the original (i.e. before discarding the explaining set) top-$k$ ranked symbols, optimally the 2nd.

## 6.2 CLEANN for Sentiment classification

We exemplify the explainability provided by our method for the task of sentiment classification of movie reviews from IMDB (20) using a pre-trained BERT model (9) that was fine-tuned for the task. The goal is to propose an explanation to the classification of a review by finding the smallest explaining set of word tokens (symbols) within the review. Figure 2 exemplifies the explanation of an input review that was classified as negative by the sentiment classification model. We compare between our method and two other baselines on finding a minimal explaining set for the classification. We see that both baselines found the word 'bad' as a plausible explanation for the negative sentiment classification, however in addition, each of them finds an additional explaining word ('pizza', 'cinema') that clearly has no influence on the sentiment in the context it appears in the review. Contrary to that, our method finds the words 'bad' and 'but' as explaining the negative-review classification. Indeed, the word 'but' may serve as a plausible explanation for the negative review at the 2nd part of the sentence, since it negates the positive 1st part of the sentence (about the pizza). Additionally, Figure 2(b) shows

| Algorithm | Review Length | | | |
|---|---|---|---|---|
| | 20 | 30 | 40 | 60 |
| Pure-Attention | 5.52 (4.34) | 8.88 (7.59) | 12.71 (10.30) | 13.14 (12.47) |
| Smart-Attention | 3.62 (2.45) | 4.63 (3.11) | 6.03 (3.34) | 6.41 (4.74) |
| CLEANN | **2.82** (1.71) | **3.48** (2.18) | **3.58** (1.83) | **3.53** (1.54) |

Table 1: Mean and standard deviation (in parenthesis) of explanation set sizes for different review lengths (number of tokens in the input sequence). Each column is calculated using 25,000 reviews. CLEANN provides explaining sets having the smallest size on average, with statistical significance tested using the Wilcoxon signed-ranks test at significance level $\alpha = 0.01$ (indicated in bold).

the corresponding attention map at the last attention layer of the model for the given input review. In addition, in Figure 2(c) we present the corresponding matrix representation of the learned graph (PAG). We can see that despite high attention values of the class token [cls] with some of the word tokens in the review (in the first row of the attention map), some of these words are found not to be directly connected to the class token in the causal graph produced by our method, and therefore may not be associated as influencing the outcome of the model.

Table 1 and Figure 3 show how the length of a review influences the explaining set size. CLEANN produces the smallest explaining sets on average, where statistical significance was tested using Wilcoxon signed-ranks test at significance level $\alpha = 0.01$. For an increasing review length it is evident that the two baselines increase their explaining set, correspondingly. Even if the additional explaining tokens are of negative/positive context, they are not the most essential and prominent word token of this kind. Contrary to that, the explaining set size produced by our method is relatively stable, with a moderate increase in the explaining set size, meaning that it keeps finding the most influential words in a sentence that dominates the decision put forward by the model.

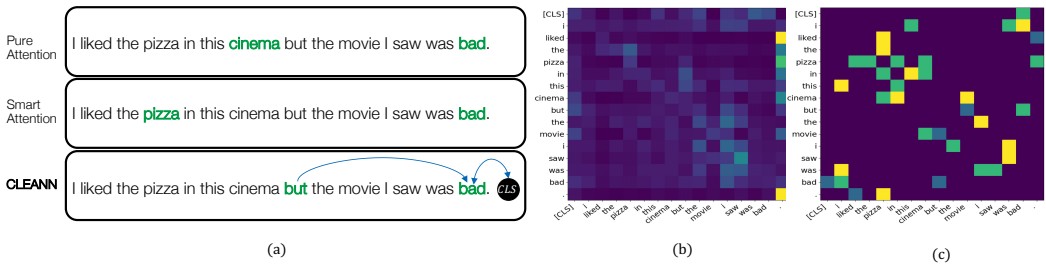

Figure 2: Comparison between the explaining sets produced by the methods given a negative review (a) Notice the implausible explaining tokens 'cinema' and 'pizza' found by the baseline methods. In contrast, in addition to the token 'bad', CLEANN found the token 'but', which is a plausible explanation as it is a negation to positive part of the sentence that precedes it. (b) Attention matrix produced for the review by the model. (c) A matrix representation of the learned graph (PAG) produced by CLEANN (green: arrow head, yellow: arrow tail, blue: circle mark).

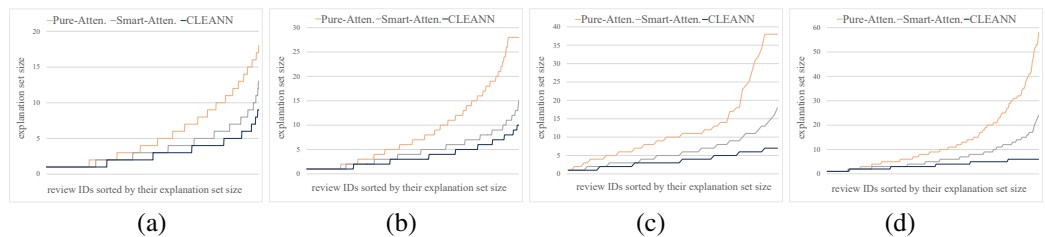

Figure 3: Comparison between CLEANN and the baseline methods on the distribution of explaining set size over the various reviews. The reviews are sorted by their corresponding explanation size on the horizontal axis. CLEANN consistently finds smaller sets for: (a) review length = 20, (b) review length = 30, (c) review length = 40, (d) review length = 60.

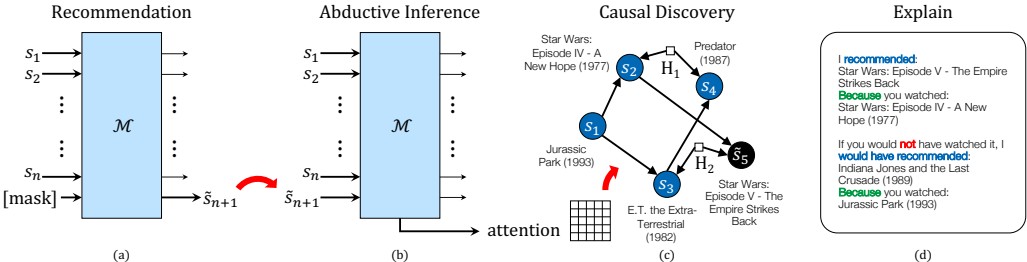

Figure 4: An overview of the presented approach. As an example, $\mathcal{M}$ is a pre-trained BERT4Rec recommender (36). (a) Given a set of $n$ movies (symbols) $[s_1, \ldots, s_n]$ the user interacted with, and the $(n + 1)^{th}$ movie masked, the recommender $\mathcal{M}$ recommends $\tilde{s}_{n+1}$. Next, in the abductive inference stage (b) the recommendation is treated as input in addition to the observed user-movie interactions, that is, input: $[s_1, \ldots, s_n, \tilde{s}_{n+1}]$, and the attention matrix is extracted. (c) A causal structure learning stage in which the attention matrix is employed for testing conditional independence in a constraint-based structure learning algorithm. The result is a Markov equivalence class. Here, as an example, a causal DAG (a member of the equivalence class) is given, where $H_1$ and $H_2$ are latent confounders. (d) A causal explanation reasoned from the causal graph.

## 6.3 CLEANN for recommendation system

We assume that the human decision process, for selecting which items to interact with, consists of multiple decision pathways that may diverge and merge over time. Moreover, they may be influenced by latent confounders along this process. Formally, we assume that the decision process can be modeled by a causal DAG consisting of observed and latent variables. Here, the observed variables are user-item interactions $\{s_1, \ldots, s_n\}$ in a session $S$, and latent variables $\{H_1, H_2, \ldots\}$ represent unmeasured influences on the user's decision to interact with a specific item. Examples for such unmeasured influences are user intent and previous recommendation slates presented to the user.

We exemplify the explainability provided by our method for the task of a recommendation system, and suggest it as a means for understanding of the complex human decision-making process when interacting with the recommender system. Using an imaginary session that includes the recommendation, we extract an explanation set from the causal graph. To validate this set, we remove it from the original session and feed the modified session into the recommender, resulting in an alternative recommendation that, in turn, can also be explained in a similar manner. Figure 4 provides an overview of our approach.

For empirical evaluation, we use the BERT4Rec recommender (36), pre-trained on the MovieLens 1M dataset (15) and estimate several measures to evaluate the quality of reasoned explanations.

**Minimal explaining set:** Figure 5(a) compares the explaining set size for the various sessions produced by CLEANN and the baseline methods. It is evident that the set sizes found by CLEANN are smaller. Figure 5(b) shows the difference between the explaining set sizes found by the baseline method and CLEANN, as calculated for each session, individually. Approximately 25% of the sessions are with positive values, indicating smaller set sizes for CLEANN, zero values shows equality between the two, and only 10% of the sessions are with negative values, indicating smaller set sizes for Pure-Attention.

**Influence of the explaining set on replacement recommendation:** Figure 6(a) compares the distribution for positions of replacement recommendations, produced by CLEANN and the baseline methods, in the *original* top-5 recommendations list. It is evident that compared to the baseline methods, CLEANN recommends replacements that were ranked higher (lower position) in the original top-5 recommendations list. In a different view, Figure 6(b) shows the relative gain in the number of sessions for each position, achieved by CLEANN compared to the baseline methods. There is a trend line indicating higher gains for CLEANN at lower positions. That is, the replacements are more aligned with the original top-5 recommendations. CLEANN is able to isolate a minimal explaining set that influences mainly the $1^{st}$ item from the original recommendations list.

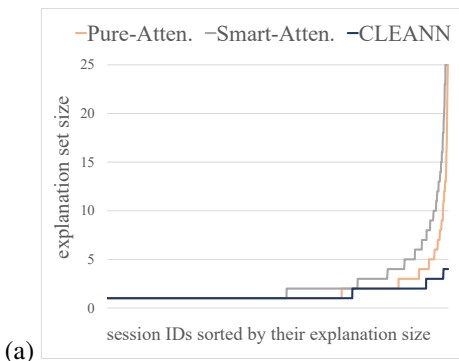
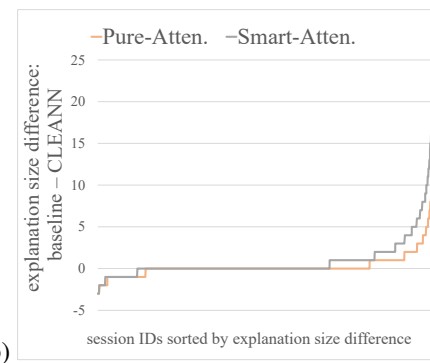

(a)           (b)

Figure 5: Explaining set size. (a) Comparison between CLEANN and the baseline methods on the explaining set size (sorted by size along the horizontal axis) for the various sessions. Set sizes found by CLEANN are smaller. (b) The difference between set sizes found by the baseline method and CLEANN, presented for each individual session. Positive values are in favor of CLEANN.

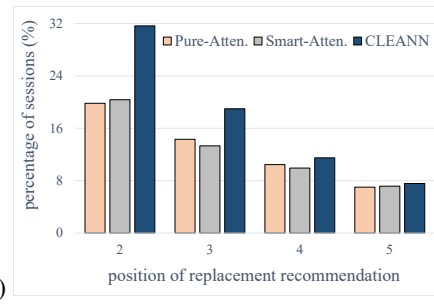
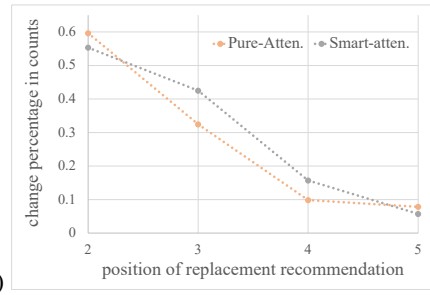

(a)           (b)

Figure 6: Positions of the replacement recommendations within the original top-5 recommendations list. (a) Position distribution of the replacements: compared to the baseline methods, CLEANN recommends replacements which are ranked higher in the original top-5 recommendations. (b) The relative gain in the number of sessions for each position, achieved by CLEANN. There is a trend line indicating higher gains for CLEANN at lower positions (replacements are more aligned with the original top-5 recommendation).

# 7 Discussion

We presented a relation between the self-attention mechanism used in the Transformer neural architecture and the structural causal model often used to describe the data-generating mechanism.

One result from this relation is that, under certain assumptions, a causal graph can be learned for a single input sequence (ABCD). This can be viewed as utilizing pre-trained models for *zero-shot causal-discovery*. An interesting insight is that the only source of errors while learning the causal structure is in the estimation of the attention matrix. Estimation is learned during pre-training the Transformer. Since in recent years it was shown that Transformer models scale well with model and training data sizes, we expect the estimation of the attention matrix to be more accurate as the pre-training data increases, thereby improving the accuracy of causal discovery.

Another result is that the causal structure learned for an input sequence can be employed to reason about the prediction for that sequence by providing *causal explanations* (CLEANN). We expect learned causal graphs to be able to answer a myriad of causal queries (24), among these are personalized queries that can allow a richer set of predictions. For example, in recommender systems, assuming that the human decision process consists of multiple pathways that merge and split, by identifying independent causal pathways, recommendations can be provided for each one, independently.

The results of this paper contribute to the fields of causal inference and representation learning by providing a relation that bridges concepts from these two domains. For example, they may lead to 1) causal reasoning in applications for which large-scale pre-trained models or unlabeled data exist, and 2) architectural modifications to the Transformer to alleviate causal inference.

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

## A Notations

In this section we provide Table 2, which describes the main symbols used in the paper.

Table 2: Notations used in the paper. The first set of symbols describes entities in the structural causal model, and the second set of symbols describes entities in the Transformer neural network.

| Symbol | Description |
|--------|-------------|
| $X_i$ | a random variable representing node $i$ in an SCM |
| $U_i$ | latent exogenous random variable $i$ in an SCM |
| $\mathbf{G}$ | weighted adjacency matrix of an SCM |
| $\mathcal{G}$ | causal graph (unweighted, directed-graph structure) |
| $\boldsymbol{Z}_i$ | output embedding of input symbol $i$, $\boldsymbol{Z}_i \equiv \mathbf{Z}(i, \cdot)$, in attention layer |
| $\boldsymbol{V}_i$ | value vector corresponding to input $i$, $\boldsymbol{V}_i \equiv \mathbf{V}(i, \cdot)$, in attention layer |
| $\mathbf{A}$ | attention matrix |
| $\mathcal{M}$ | Transformer neural network |
| $\mathbf{W}_V, \mathbf{W}_{QK}$ | learnable weight matrices in the Transformer neural network |

## B Finding Possible Explanations from a Causal Graph

This appendix includes the pseudo-code for the CLEANN (**C**ausa**L E**xplanations from **A**ttention in **N**eural **N**etworks) algorithm and explanatory figures. By abduction, we replace the masked symbol with a predicted one, creating an imaginary sequence. We then feed-forward this sequence and extract the resulting attention matrix (Algorithm 1-lines 2–4). Using this attention matrix we learn a causal graph as discussed in Section 3.2 (Algorithm 1-lines 5–7).

Given a causal graph, various "*why*" questions can be answered (26). We follow (38) explaining a prediction using the symbols from the input sequence. That is, provide the minimal set of symbols that led to a specific prediction and provide an alternative prediction. To this end, we define the following.

**Definition 1** (PI-path). *A potential influence path from $A$ to $Z$ in PAG $\mathcal{G}$, is a path $\Pi(A, Z) = \langle A, \ldots, Z \rangle$, such that for every sub-path $\langle U, V, W \rangle$ of $\Pi(A, Z)$, where there are arrow heads into $V$ and there is no edge between $U$ and $W$ in $\mathcal{G}$.*

Essentially, a PI-path ensures dependence between its two end points when conditioned on every node on the path (a single edge is a PI-path).

**Definition 2** (PI-tree). *A potential influence tree for $A$ given causal PAG $\mathcal{G}$, is a tree $\mathcal{T}$ rooted at $A$, such that there is a path from $A$ to $Z$ in $\mathcal{T}$, $\langle A, V_1, \ldots, V_k, Z \rangle$, if and only if there is an PI-path $\langle A, V_1, \ldots, V_k, Z \rangle$ in $\mathcal{G}$.*

**Definition 3** (PI-set). *A set $\boldsymbol{E}$ is a potentially influencing set (PI-set) on item $A$, with respect to $r$, if and only if:*

1. *$|\boldsymbol{E}| = r$*

2. *$\forall E \in \boldsymbol{E}$ there exists a PI-path, $\Pi(A, E)$ such that $\forall V \in \Pi(A, E), V \in \boldsymbol{E}$*

3. *$\forall E \in \boldsymbol{E}$, $E$ temporally precedes $A$.*

An example for identifying PI-sets is given in Figure 7. Although a PI-set identify nodes that are conditionally dependent on the prediction, it may not be a minimal explanation for the specific prediction. That is, omitting certain symbols from the session alters the graph and may render other items independent of the prediction. Hence, we provide an iterative procedure ("FindExplanation" in Algorithm 1-lines 10–18), motivated by the ICD algorithm (30), to create multiple candidate explanations by gradually increasing the value of $r$. See an example in Figure 8. Note that from conditions (1) and (2) of Definition 3, $r$ effectively represents the maximal search radius from the prediction (in the extreme case, the items of the PI-set lie on one PI-path starting at the prediction). The search terminates as soon as a set qualifies as an explanation (Algorithm 1-line 17). That is, as soon as a different prediction is provided. If no explanation is found (Algorithm 1-line 19) a higher value of $\alpha$ should be considered for the algorithm.

---
**Algorithm 1:** CLEANN: CausaL Explanations from Attention in Neural Networks
---

**Input:**
    $s$: a sequence $s = [s_1, \ldots, s_n, \langle mask \rangle]$
    $\mathcal{M}$: a pre-trained attention-based model
    $\tilde{s}_{n+1}$: prediction by $\mathcal{M}$
    $\alpha$: significance level for CI-testing

**Output:**
    $E$: an explanation set for the prediction
    $\tilde{s}'_{n+1}$: an alternative prediction (after masking the effect of the explanation)
    $\mathcal{G}$: a causal graph of $[s_1, \ldots, s_n, \tilde{s}_{n+1}]$

**1**   **Function** `Main`($s$, $\mathcal{M}$, $\tilde{s}_{n+1}$):
**2**      $\tilde{s} \leftarrow [s_1, \ldots, s_n, \tilde{s}_{n+1}]$                 ▷ `replace mask with prediction`
**3**      forward pass: $\mathcal{M}(\tilde{s})$
**4**      get $\tilde{\mathbf{A}}$, the attention matrix in the last layer of $\mathcal{M}(\tilde{S})$
**5**      define correlation matrix $\boldsymbol{\rho}_{\tilde{S}} : \boldsymbol{\rho}_{\tilde{S}}(i,j) = \mathbf{C}(i,j)/\sqrt{\mathbf{C}(i,i)\mathbf{C}(j,j)}$, where $\mathbf{C} = \tilde{\mathbf{A}}\tilde{\mathbf{A}}^\top$
**6**      define $\mathrm{Ind}(\boldsymbol{\rho}_{\tilde{S}})$ : a conditional independence test based on partial correlation
**7**      $\mathcal{G} \leftarrow$ `CausalDiscovery` $(\mathrm{Ind}(\boldsymbol{\rho}_{\tilde{S}}), \alpha)$
**8**      $E, \tilde{s}'_{n+1} \leftarrow$ `FindExplanation` $(\mathcal{G}, \tilde{s}_{n+1}, \mathcal{M}, s)$
**9**      **return** $E$, $\tilde{s}'_{n+1}$, $\mathcal{G}$

**10**   **Function** `FindExplanation`($\mathcal{G}$, $\tilde{s}_{n+1}$, $\mathcal{M}$, $s$):
**11**      create $\mathcal{T}$: a PI-tree for $\tilde{s}_{n+1}$ given $\mathcal{G}$
**12**      **for** $r$ *in* $\{1, \ldots, n-1\}$ **do**
**13**          create $\mathcal{E}$, subsets of $\mathbf{Nodes}(\mathcal{T}) \setminus \tilde{s}_{n+1}$ such that $\forall E \in \mathcal{E}$, $E$ is a PI-set on $\tilde{s}_{n+1}$ in $\mathcal{G}$
             with respect to $r$.
**14**          **for** $E \in \mathcal{E}$ **do**
**15**              $s' \leftarrow s \setminus E$
**16**              $\tilde{s}'_{n+1} \leftarrow \mathcal{M}(s')$              ▷ `get an alternative prediction`
**17**              **if** $\tilde{s}'_{n+1} \neq \tilde{s}_{n+1}$ **then**
**18**                  **return** $E, \tilde{s}'_{n+1}$        ▷ `return immediately ensuring smallest`
                    `explanation set`
**19**      **return** $\emptyset, \emptyset$      ▷ `no explanation found at the current significance level`

## C   Proofs

In this section we provide proofs for the results described in Section 4.2. Let $(\Omega_n, \mathcal{F}_n, P_n)$ be an SCM-consistent probability space, and $\mathcal{M}$ trained on sequences from this probability space, complying with Assumption 1.

**Theorem 1.** *If $s$ is a sequence from an SCM-consistent probability space used to train $\mathcal{M}$, $\mathcal{G}_s$ is the causal graph of the SCM from which $s$ was generated, and $\mathbf{A}$ the attention matrix for input sequence $s$ in the deepest layer, then $\mathcal{G}_s$ is in the equivalence class learned by a sound constraint-based algorithm that uses covariance matrix $\mathbf{C}_{\mathbf{Z}} = \mathbf{A}\mathbf{A}^\top$ for partial correlation CI-testing.*

*Proof.* From Assumption 1 it follows that an output embedding, in the deepest attention layer, for each symbol is trained to predict the input symbol. It follows that the output embeddings of the deepest layer are one-to-one (injective) representations of the input symbols. Thus, the causal relations among the input symbols can be cast on their correponding output embeddings. That is, a causal graph over the output embeddings is similar to causal graph over the input symbols $\mathcal{G}_{\mathbf{Z}} = \mathcal{G}_s$. In Section 4.1 we showed that self-attention in the Transformer architecture can be viewed as an SCM over the output embedding, and the covariance matrix, $\mathbf{C}_{\mathbf{Z}}$, can be computed from the attention matrix (Equation 9). A conditional independence test between output embeddings can then be defined to calculate partial correlations from $\mathbf{C}_{\mathbf{Z}}$. Finally, a sound constraint-based algorithm (35) that uses $\mathbf{C}_{\mathbf{Z}}$ for partial correlation CI-testing, recovers an equivalence class of $\mathcal{G}_{\mathbf{Z}}$.      □

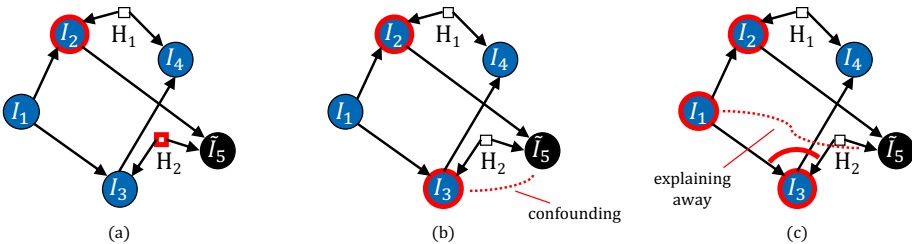

Figure 7: An example for a PI-set on $\tilde{I}_5$. Latent nodes are represented by squares. Nodes included in the PI-set are circled with a thick red line. Indirect dependency is marked with a dashed red line. (a) The smallest PI-set for a fully observed DAG is the Markov blanket $\{I_2, H_2\}$. However, $H_2$ is latent. (b) Nodes $I_2, I_3$ are included in the PI-set as they have direct effect on $\tilde{I}_5$, where $I_5$ depends on $I_3$ due to the latent confounder $H_2$. However, $I_3$ is a collider, hence including it in the PI-set results in explaining-away and the PI-set is not complete. (c) A complete PI-set $\{I_1, I_2, I_3\}$. Including $I_3$ creates an indirect dependence between $I_5$ and $I_1$ (a PI-path), which requires including $I_1$ as well. Note that the distance from $I_5$ to $I_1$ is two edges in the graph over observed variables.

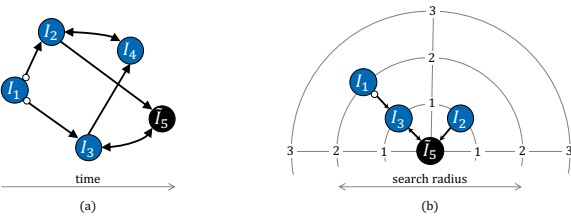

Figure 8: An example for (a) a causal graph (PAG), and (b) the nodes that influence the prediction $\tilde{I}_5$ having a PI-tree structure, depicted on coordinates where the radius axis is the distance from the prediction. In Algorithm 1-line 12, the algorithm iterates over radius values, starting with 1. In the first iteration, the sets $\{I_2\}$ and $\{I_3\}$ are tested if any of them qualifies as an explanation. If both sets fail, the search radius increases to 2 and the sets $\{I_2, I_3\}$ and $\{I_1, I_3\}$ are tested (set $\{I_2, I_3\}$ is tested first as it members have smaller average distance from $\tilde{I}_5$). If both fail, $r = 3$ and the last tested set is $\{I_1, I_2, I_3\}$.

**Corollary 2.** *Nodes $X_i$ and $X_j$ are d-separated in $\mathcal{G}_{\boldsymbol{Z}}$ conditioned on $\boldsymbol{X}' \subset \boldsymbol{X}$ if and only if symbols $s_i$ and $s_j$ are d-separated in $\mathcal{G}_{\boldsymbol{s}}$ conditioned on $\boldsymbol{s}'$, a subset containing symbols corresponding to $\boldsymbol{X}'$.*

*Proof.* This result follows directly from Theorem 1 and faithfulness assumption (Definition 2). □

