# OpenReview forum: "Causal Interpretation of Self-Attention in Pre-Trained Transformers"
_NeurIPS.cc/2023/Conference — NeurIPS 2023 poster_

### Official Review · Reviewer_unWZ · 2023-06-25

**Soundness:** 2 fair
**Presentation:** 2 fair
**Contribution:** 2 fair
**Rating:** 5
**Confidence:** 4

**Summary:**

The paper establishes a formal link between a (pre-trained) self-attention layer and a causal graph underlying a sequence of symbols. Specifically, it shows that self-attention learns to represent pairwise associations for which there exists a linear-Gaussian Structural Causal Model that has the exact same pair-wise associations between its nodes. The paper then proceeds to derive an algorithm for recovering the causal structure of the SCM solely from the weights of a pre-trained attention layer

**Strengths:**

The Attention-Based Causal-Discovery (ABCD) method, which is developed for learning an equivalence class of the causal graph that underlies a given input sequence is reasonable. It relies on partial correlarion CI-testing, which is sound.
Then the paper proceeds to apply ABCD to reasoning about predictions by generating causal explanations from pre-trained self-attention models, such as BERT (8). Specifically, the algorithm produces explanations through selecting a subset of the input symbols that are claimed to be solely responsible for BERT prediction. ABCD is shown empirically to be much stronger than the common approach of using the attention matrix to learn about input-output relations for providing explanations.

**Weaknesses:**

The Attention-Based Causal-Discovery (ABCD) method relies on an unsupervised training assumption. This restricts applicability to only models of BERT type, you cannot deal e.g. with decoder-only models.

In addition, it is extremely troubling that the authors do not perform any comparative evaluation against standard methods in the field. They limit their examination to standard self-attention and a variant of it. Even worse, they have avoided use of any standard benchmark in the field.

Eventually, it is not clear how useable the generated causal explanations are, and how easy it is to exploit them in an application setting.

**Questions:**

1. I want to see a serious reconsideration of all the empirical evidence the paper provides.
2. How can the causal explanations be used in an easy and/or automated setting?

---

> ### Author Rebuttal · Authors · 2023-08-09
>
> We sincerely thank you for your review, and the important questions. We hope that our response below makes this paper's contribution clearer, and mitigates the concerns you raised.
>
> **Re: weaknesses.**
> We wish to clarify that the main contribution of the paper is twofold. The first is a link between the unknown causal model governing the input symbols/token and a simple linear-Gaussian SCM over the embeddings calculated in the deepest self-attention layer. To the best of our knowledge, this is the first link made between pre-trained self-attention-based models and SCM (Figure 1). The second part of the contribution is a method that recovers a causal structure for a specific input sequence in a zero-shot manner. Generally in causal discovery, a large number of samples generated from a single causal models are used to recover this single causal model. In this contribution, each data sample is generated by a different causal model (Definition 6, Section C in the supplementary material). To the best of our knowledge, this is the first method for this setting and the first method that learns causal graphs from pre-trained self-attention. As you pointed out, this paper proposes a solution for self-attention and not for cross-attention (or decoder-only models). We note that this paper proposes a novel and non-trivial causal interpretation to self-attention, and since a similar treatment for cross-attention is not an incremental extension we leave it for future work.
>
> Regarding benchmarks, currently there is no known benchmark that contains a set of sequences (data samples) and their corresponding causal graphs; each sample has its own graph. Nevertheless, we quantitively evaluated explanations deduced from the learned causal graphs, and demonstrated their superiority over baselines that use the values of self-attention. This quantitively measures the usefulness of causal structures learned using the novel link established by ABCD.
>
> Regarding the usability of explanations and their ease of application in an automated system, here are three ways. 1) one example is improving training by penalizing/rewarding the explaining set, which can be traced-back to the attention weights, of an incorrect/correct prediction. This can be done for each sample in the training data independently. 2) In the paper we demonstrated the usefulness of ABCD for automatically creating causal explanations for the BERT4Rec [28] recommender system by proposing the CLEANN algorithm (Section B). An explanation to why a specific recommendation was given to a human in a tangible way can lead to greater trust in the recommendation, and greater human engagement with the system. 3) Finally, (in the recommender system setting) understanding the human decision process (causal graph over user-item interactions) can be used to enhance recommendations (beyond 'linear' recommendations). For example, by identifying independent causal pathways (some may pause in time, before others), and creating recommendations for each pathway, thereby enriching recommendations.
>
> **Re: Questions.**
> 1. Please see the second paragraph in 'Re: weaknesses'.
> 2. Please see the third paragraph in 'Re: weaknesses'.
>
> We are confident that addressing the concerns you raised will improve the clarity of this paper and how it can promote future work in bridging Transformers and causality. We hope that our answers will allow reexamination of the contribution of this paper.

---

> > ### Comment · Reviewer_unWZ · 2023-08-12
> >
> > I would like to thank the authors for the many clarifications. In view of these responses I have increased my score

---

### Official Review · Reviewer_qdMo · 2023-07-06

**Soundness:** 3 good
**Presentation:** 2 fair
**Contribution:** 3 good
**Rating:** 6
**Confidence:** 3

**Summary:**

The ABCD paper starts off by motivating the need for understanding the complex dependencies that underly current Transformer architectures, pointing to causality as the missing puzzle piece. Following brief introductions of both SCMs and Self-attention, the authors then provide covariance matrix derivations for both linear Gaussian SCM and attention such that they are able to equate them eventually. By introducing another assumption on how the attention was trained, they propose a theorem that provides a clear connection between causal discovery and attention in that one can make CI tests based on the attention matrix essentially. Finally, the authors propose a complete pipeline for going from e.g. a pre-trained Transformer model to causal explanations based on their ABCD approach computed from the last attention layer. As put by the authors, this can be considered as a "one-shot" causal discovery paradigm.

**Strengths:**

1. A unique perspective and valuable contribution to the causal discovery literature in that it expands the horizon on what kind of connections can be established between existing models to foster new understanding of said models. The derivation of the covariances and then equating them is both simple and effective, and then Thm.1 seems like a nice result for understanding the final correspondence between attention and CI-testing for constraint-based causal discovery.

2. Presenting a complete pipeline is a great bonus (since it is a difficult task) and shows an immediate practical approach to the previously presented theory. The explanation approach is convincing and practically applicable since the theory justifies the use of the attention matrix.

**Weaknesses:**

1. While the causality assumptions Markov and faithfulness can be considered standard, as soon as moving to the realm of data that nowadays Transformers are confronted with we run the risk of violating them. Furthermore, the assumption of a linear Gaussian SCM is quite restrictive and should be highlighted more, especially under consideration of Transformers. Finally, Assumption 1 right before Theorem 1 is only vaguely comprehensible IMHO. I'd wish for the authors to be more specific about what property is required for the deepest attention layer such that the equating to CB-CD works. This could be done through formalization of that assumption or if not possible, then through an example.

2. Thm.1 is incomplete writing-wise i.e., sentence cuts off at end of page 4. Starting from p.4 the paper generally lend the impression of being not as carefully written as the pages before. Please consider a careful readthrough post-submission.

3. Missing discussion of explanation methods based on causal graphs. Two very different approaches include Causal Shapely Values by Heskes et al. 2020 and Causal Explanation of SCM by Zečević et al. 2021. The first is a rather classical approach, where numerical attributions are being computed and in Sec.5 they present an approach for causal chain graphs (a special case like your Linear-Gaussian case), and the second discusses a traversal algorithm for generating immediate textual representations based on more general SCM (although only showcased for a linear SCM setting).

**Questions:**

1. Can you comment on Theorem 1 as mentioned in the weaknesses section?

2. The assumption of linear Gaussian SCM can severely limit the method. What was the reasoning behind the assumption?

3. What property is required for the deepest attention layer such that the equating to CB-CD works.

Overall, a nice contribution but not devoid of weaknesses. I will be willing to raise my score based on the rebuttal.

**Limitations:**

No concerns here.

---

> ### Author Rebuttal · Authors · 2023-08-09
>
> We wish to thank you for you detailed review, positive feedback, and identifying weakness that addressing them significantly improves this paper.
>
> **Re: Weaknesses.**
> 1. **(a) Re: Causal Markov and faithful assumptions.** The causal Markov and faithfulness assumptions are required for constraint-based causal discovery. We will clarify this just before their definition in the preliminaries section by adding in line 61 'In this paper, we learn the causal graph, described by the non-zero elements of $\mathbf{G}$, using constraint-based causal discovery approach. It requires the causal Markov and faithfulness assumptions.'. Nevertheless, they are not required for the link established between SCM and self-attention in Transformers. We will clarify that a violation of these assumptions will hinder the accuracy of the causal discovery algorithm (third paragraph of Section D: 'Limitations of ABCD' in the supplementary material). We will move Section D to the main paper using the 1 extra-page provided for the final camera-ready version.
> **(b) Re: linear Gaussian SCM.** We wish to clarify that this is not a restriction (assumption) but rather a result. We showed in section 3.1 that self attention can be viewed as a mechanism that estimates values of a linear-Gaussian SCM nodes. Specifically, Equation 4 describes the total effect one variable has on another one via all directed paths of a linear-Gaussian SCM. This is similar to the attention one token is given by another token. We then show in Equation 9 that the covariance over endogenous variables of the linear-Gaussian SCM can be computed using the attention matrix. The Transformer learns to 'disentangle' complex relations between input symbols into linear relations in the last attention layer (under Assumption 1).
> **(c) Re: Assumption 1.** In the supplementary material in Section C (Proofs), we provided a formal definition to the probability space (Definition 6) on which the assumption relies. In addition, since it is referred to in the proof of Theorem 1, a motivation for it is given. We will move the formal definition of the probability space and the accompanying explanations to the main paper using the extra 1 page available for the camera-ready version. We will also explain that common pre-training of self-attention (e.g., Mask Language Model, MLM in BERT) validate this assumption. In addition, we will explain that commonly in an unsupervised training of Transformers, the output embedding of a token alone (ignoring the output embeddings of other tokens) is used to predict the input token.
> 2. We regret that the Theorem cuts off and sincerely apologize for that. The full Theorem 1 along with more a formal description is in Section C of the supplementary material. We will correct this along with other clarities you spotted for the camera-ready version, based on the supplementary material.
> 3. We will discuss both papers as suggested with respect to the CLEANN algorithm.
>
> **Re: Questions.**
> 1. Theorem 1, and other related details will be corrected based on Section C of the supplementary material (as addressed in our response to 'weaknesses 2'.
> 2. The linear-Gaussian SCM is not an assumption. It is a result. We will clarify this in the paper as discussed in Section C in the supplementary material. In section 3.1 we showed that self-attention can be viewed as a mechanism that estimates the total effect one node has on another node in a Linear-Gaussian SCM, and used this to calculate the covariance (Equation 9). We addressed this question in more detail in our response to 'weaknesses 1.b'.
> 3. The output embeddings of the deepest layer are one-to-one (injective) representations of the input symbols (Section C, proof of Theorem 1). That is, the input token should be able to be predicted using only its corresponding output embedding of the deepest layer. This commonly the optimization goal in Transformer pre-training (e. g., MLM for BERT). We provide more details in our response to 'weaknesses 1.c'.
>
> We would like to sincerely thank you for the questions and suggestions that allowed us to improve the overall quality and clarity of this paper.

---

> > ### Comment · Reviewer_qdMo · 2023-08-12
> > **Thank you for the response**
> >
> > I would like to thank the authors for their detailed response. I am overall positive about the paper and would like to keep my score.

---

### Official Review · Reviewer_75HH · 2023-07-06

**Soundness:** 3 good
**Presentation:** 2 fair
**Contribution:** 3 good
**Rating:** 7
**Confidence:** 3

**Summary:**

In this work, authors, interpret self attention in transformers as linear gaussian SCM over output embeddings.
SCM over input embeddings is same as SCM over output embeddings, since deepest attention layer is trained to predict input symbol.
Existing conditional independence based testing algorithm is applied on the weights of the deepest attention layer of the pretrained transformer in order to estimate causal structure.
Furthermore, the authors demonstrate the applicability of the proposed approach in deducing causal explanations for attention based recommendations.

**Strengths:**

Authors proposed novel way of viewing self attention in transformers as weight matrix of linear gaussian SCM
Pretrained model is used to obtain causal structure, which would be of great use to gain causal perspectives for existing model
Estimating causal structure from the attention layer and using it for causal explanantions is promising.

**Weaknesses:**

Theorem1, in main paper has few words missing, however, it's available in supplementart material

**Questions:**

In the paper, the covariance matrix of value is assumed to be identity using central limit theorem, however, in practice, it wouldnot be identity as a result the graph learnt wouldnot be faithful. This will act as one more reason of inaccuracy. Is it possible for you to throw some light on this ?

**Limitations:**

1) The accuracy of the proposed method as explained by the authors is dependent on the prediction of the transformers and the conditional independence testing accuracy
2) Authors assumed learnt graph will be faithful, since exogeneous variables V under central limit theorem will have identity covariance, however in practice the faithfullness assumption will be violated, this will act as one more reason of inaccuracy

---

> ### Author Rebuttal · Authors · 2023-08-09
>
> First, we wish to thank you for your review, positive feedback, and questions that allowed us to further improve this contribution.
>
> **Re: Weaknesses.**
> Thank you for spotting the cut-off sentence, and finding it complete in the supplementary material. We will correct that.
>
> **Re: Questions.**
> Each column of $\mathbf{W}_V$ represents an axis on which the input is projected resulting an instance for the exogenous values of the SCM (dot-product between embedding of token $i$ from the previous layer with column $j$ of $\mathbf{W}_V$ is the element $(i, j)$ of the values matrix). Thus, each column of the values-matrix, $\mathbf{V}$, is treated independently, and the covariance is a diagonal matrix. Also, STD is 1 on the diagonal and scaled within $\mathbf{A}$ as can be seen by parameter $\Lambda$ that scales the STD in SCM in Equation 9. Thus, we consider the identity covariance matrix. From the SCM perspective, each column of $\mathbf{V}$ is a unique context (instance of the exogenous variables) for which $\boldsymbol{X}$ values are calculated. The $\boldsymbol{X}$ values is a column in the self-attention output $\mathbf{Z}$. Each column corresponds to a specific context, and the concatenated columns into matrix $\mathbf{Z}$ is used for further processing by the Transformer (see Figure 1).
>
> **Re: Limitations.**
> 1. As you indicated, we described the limitations in the supplementary material. For improving the clarity we will move Section D, 'Limitations of ABCD' to the main paper using the extra 1 page available for the camera-ready version.
> 2. We will add to the limitations section that violation of faithfulness, which might happen in real-world cases, will cause errors in the constraint-based causal discovery algorithm. Nevertheless, this does not reflect a violation for the link we derived between self-attention and SCM.
>
> Addressing the points in your review improves this contribution, and we hope that in light of our clarifications and answers you will consider updating the overall score of this contribution.

---

> > ### Comment · Reviewer_75HH · 2023-08-21
> >
> > I would like to thank the authors for the many clarifications. In view of these responses I have increased my score

---

### Official Review · Reviewer_wsNj · 2023-07-25

**Soundness:** 3 good
**Presentation:** 3 good
**Contribution:** 3 good
**Rating:** 5
**Confidence:** 3

**Summary:**

The paper proposes Attention-Based Causal Discovery (ABCD), a causal interpretation for the attention matrix of transformers (e.g., LLMs like BERT). The authors suggest CLEANN, an algorithmic approach that first finds a causal structure of the input symbols (e.g., tokens (words) of a sentence as input to BERT). This causal structure is unique to each input (sentence). This can then be used to get causal explanation for different taks (in their experiments classification or recommendation) based on the representations of the input sentences (e.g., the word "but" and "bad" in the input sentence caused the negative sentiment classification).

**Strengths:**

Thank you for your submission. I enjoyed reading your paper.

Clarity & Presentation: I found the paper to be clearly structured and written. On clarity and presentation, I would like to positively highlight the introduction to each (sub) section gave an orientation that helped to clearly guide the reader, what the next chapter is about and how it fits in the bigger picture. The figures (e.g., Fig2a, Fig 4) helped a lot to understand the method/pipeline.

Novelty: I found the causal interpretation and its application to causal explanation of predictions apealing and novel. Finding a causal explanation to downs-stream tasks of LLMs for example, can be very helpful. However, I am not an expert in the field of LLMs and I do not feel qualified to fully assess the novelty and hence the contribution of the work in the general field.

Evaluation: I found the experimental section clearly structured and interesting to read.

**Weaknesses:**

I have a few comments for improvements.

* Section 5, Table 1: What do the bold numbers indicate, what the +- (i.e., over how many runs did you do it, were these runs of CLEANN or different data samples)? This would be helpful to comment in the caption. Am I right in understanding tha CLEANN does yield lower average explanations, but the numbers suggest that it is not a significant? Over how many runs did you do it?

* Section 5, Figure 2 (b, c): I find it hard to understand and interpret Figure 2(b,c) despite your explanation in ln. 255ff. Also, the labels are very small and hard to read.

* Section 5, Figure 3: I am missing an explanation/interpretation of Figure 3 in the text. Did I overlook it?

* I am missing a related work section.

Minor:
* Section 1: While I enjoyed reading the empirical evaluation, I found the causal explaining application could be stronger highlighted in the introduction.

* Section 2.1, line 67: While I appreciate the introduction to SCMs, I was a bit confused about the Causal Markov and Faithful properties at the end of the section. They seem very disconnected of the rest of the text. It would help, if there would be a short explanation, why they are stated / needed for the paper.

* Section 2.2 is very brief. In the light of Section 3 (ln. 100 ff.), where the self-attention mechanisms is further detailed, it makes sense. However, it may be helpful to just point to that at the end of the section 2.2.

* Section 5: Caption of Table 1 - by review length, you mean the length of the sentence, right? It would have helped me  to point that out.

* Section 5: I find it weird that Fig. 6 is discussed before Fig. 5.  A natural way would be to discuss them the other way around.

* I got a bit confused by the difference between ABCD and CLEANN. Could you kindly clarify?

Typos:
* ln 222 "(1) In addition, it" ->  "(1) It"
* ln 262 "influences on the " -> "influences the"



**Questions:**

I have a few questions:

* Section 2.1, ln. 50, Definition of an SCM. I have previously seen an SCM being defined often $M = (F  P(U))$, or $M = (X, U , F)$, but I have not seen both U or P(U). Is yours the proper formal definition?

* Section 2.1, ln. 59, your understanding of a linear-Gaussian SCM is the same as an Additive Noise Model (ANM)?

* Section 5, Fig. 6: I do not understand the x-axis. Could you explain this again?

* Are there other applications of ABCD beyond providing causal explanations?


**Limitations:**

Section 6 Discussion:  ln. 305 mentions "under certain assumptions", but these assumptions are not re-stated here. It would be helpful to have them restated. In this regard, I am missing a discussion of the limitations regarding the (causal) assumptions.

I am also wondering, how robust your causal explanations are? I believe, when providing causal explanations to individuals, it is important to discuss any limitations regarding the sensitivity to assumptions and to comment the potential negative societal impact, if assumptions are not met.

---

> ### Author Rebuttal · Authors · 2023-08-09
>
> We wish to thank you for the very detailed review and your positive feedback. Also, thank you for your comments for improvements that mitigate potential weaknesses. Addressing them as suggested strengthens the contribution and makes it clearer.
>
> **Re: Your comments for improvements (weaknesses).**
> * Re. Section 5, Table 1.: We will add the following clarification to the caption and in the text. The bold numbers indicate statistical significance with Wilcoxon signed-ranks test at significance level $\alpha=0.01$ (see also sorted pairwise difference in Figure 3). They are also statistically significant with the single-sided $t$-test. In both tests the $p$-value is very small. Mean and standard deviation are calculated over 25000 samples. We will remove the $\pm$ sign and place standard deviation values in parenthesis.
> * Re. Section 5 Figure 2 (b, c): We will increase the size of depicted matrices in Figure 2 (b ,c). In addition, we will clarify in the caption that these illustrate that high attention values do not necessarily translate to direct causal relation as in the case for CLS highest attention to the token 'was' (2.b) whereas it is disconnected from it in the causal graph (0 value in graph matrix in 2.c).
> * Section 5 Figure 3:  Thank you for spotting this. Line 257 should have been "Table 1 and Figure 3 show …", instead of "Table 1 shows". Figure 3 is given to demonstrate the significance of the results summarized in Table 1 (it is clear from Figure 3 that there is statistical significance when using the Wilcoxon signed-ranks test).
> * Related work is Section E in the supplementary material. We will move it to the main paper.
>
> **Re: Minor comments.**
> * As suggested we will add the causal explaining application at the end of the first introduction paragraph (line 24).
> * We will replace line 61, starting with ('In this paper we …'), with 'In this paper, we learn the causal graph, described by the non-zero elements of G, using constraint-based causal discovery approach. It requires the causal Markov and faithfulness assumptions.'. This will make a clearer connection to the assumptions. Thanks.
> * We will clarify at the end of section 2.2 that in Section 3 we will further detail self attention as required for establishing a link to SCM.
> * 'review length' means total number of tokens. we will improve Table 1 caption as suggested.
> * We will fix the ordering of Figures 5 and 6 as suggested.
> * ABCD is the approach that links SCM and self-attention, and provides means to learn the causal structure from pre-trained transformers. CLEANN is an algorithm that firstly employs ABCD to learn a causal structure. It then iteratively traverses the learned structure and searches for an explanation for the Transformer's output, which involves forward passes of the neural network to test the explanation.
>
> **Re: Typos.**
> Thank you for identifying these typos. We will correct them as well as proofread the final version.
>
> **Answers to the question.**
> * Definition of SCM: Since we are interested in the covariance of endogenous variables, we follow the definition of probabilistic SCM by Pearl (Causality, 2009, definition 7.1.6). This is also the definition of SCM generally used by E. Bareinboim (e.g., Xia et al., 2021, Definition 1. Reference [10] in the supplementary material).
> * The noise in the Linear-Gaussian SCM we used is additive and is normally distributed.
> *  The X-axis in Figure 6: We ordered the sessions (data samples) on the X-axis in an order obtained by sorting the values of the Y-axis (e.g., explaining set size per session). This way, the plots are monotonically increasing. We will clarify the label. By sorting the values, we provide a description of the empirical cumulative distribution function (CDF). Figure 6-(a) describes how the explanation sizes are distributed. Figure 6-(b) describes how the difference between CLEANN and other baselines is distributed.
> * In essence, ABCD enables zero-shot causal discovery. Inferring the causal relations solely from observational data is important in many sciences (e.g., healthcare, economics, physics, chemistry). In this paper we demonstrated one practical use. It is readily practical for currently available Transformers. We expect that future applications, interested in recovering underlying causal structures in a zero-shot manner, may train attention-based models and use ABCD. For example, understanding the health condition and its direct causes for a specific user, after training on a large dataset of observations for humans. Another possible example is pretraining a BERT model on sequences of amino acids (proteins), and inferring various relations that may lead to insights on the specific protein's folding mechanism. These examples are not part of the paper and given here just for describing potential use of ABCD.
>
> **Re: Limitations.**
> We discussed the limitation of ABCD in Section D in the supplementary material. We will move it to the main paper (using the extra 1 page available for the camera ready version). We will refer that the violation of the assumptions leads to errors in constraint-based causal discovery (as discussed in the third paragraph of Section D). Also, as suggested, we will restate them in the discussion.
>
> Finally, in light of your initial positive review and considering our answers we hope you will reconsider your initial overall score.

---

> > ### Comment · Reviewer_wsNj · 2023-08-17
> >
> > Dear authors, apologies for the late reply. Thank you very much for your detailed response.
> > In the light of the respones and the concrete suggestions on how to implement the changes in the paper, I have reaised my score.

---

### Decision · Program_Chairs · 2023-09-21

**Decision:**

Accept (poster)

**Comment:**

The paper presents a causal interpretation for the attention matrix of transformers. While all reviews agree that this interesting, the authors have to highlight and discuss much more the assumptions made, in particular, the main assumption of causal Markov and faithfulness. Indeed, they are mentioned but it is not stressed that this simplified the whole story to independence testing. This is important, as there a known connection between transformers (self attention) and probability densities. There are even probabilistic attention layers. This discussion is very much required. Apart from this, I agree with the overall positive assessment of the reviewers.